# In Situ Visual Feedback for Learning Hand Gestures in VR

Category: Research

## ABSTRACT

We investigate *In Situ* visual feedback, on or in the hand, to learn hand gestures more precisely in virtual reality environments. This feedback helps users understand if any of their fingers are incorrectly positioned during a gesture, and enables hitting a smaller, more precise "target" in pose space. In Situ feedback can be used to teach gestures in a tutorial mode, or can appear automatically when the user gets close enough to a known hand pose, serving both as an autocompletion hint (i.e., if the user terminates the gesture, they know with confidence how it will be interpreted) and as a feedforward hint (i.e., the user can adjust their pose to be closer to the ideal before terminating, when precision is important). We present four variants of In Situ feedback, which were compared in a first study in virtual reality involving 12 users wearing a Meta Quest 2 headset. The most promising variant, Puppet, was evaluated in a second study with 20 users, and enabled greater precision than a static grid, and was preferred by most users.

**Index Terms:** Human-centered computing—Virtual reality; Human-centered computing—Graphical user interfaces

## 1 INTRODUCTION

Hand tracking is now common in XR (virtual and augmented reality), for example in games for the Meta Quest 2, or business applications with the Microsoft Hololens 2. Hand tracking is also sometimes used without headsets, both in workstations [25, 28] and in public displays such as in shopping centers. Gestural interaction with bare hands has advantages over input devices like keyboards, mice, and hand-held controllers: bare hand gestures (1) do not require the user to pick up hardware devices, which may be hard to find when wearing an opaque headset or may need to be recharged or may require learning how to use; (2) afford the user more freedom to stand or walk during interaction; (3) are more easily intermixed with the grabbing and manipulation of other physical objects; and (4) there are more degrees of freedom available with bare hands (position, orientation, individual fingers) than with common input hardware devices. Once learned, gestures allow for fast execution of commands with no need for menus or widgets that would occupy space.

Status quo interfaces often reveal available gestures through visual aids (static images or animations) that are separate from the user's hands. These visual aids have the disadvantages of occupying screen space, requiring the user to move their eyes between the visual aid and their hand, and also make it difficult to achieve precisely the same gesture with all fingers positioned correctly. Learning to reproduce gestures precisely helps to avoid subsequent errors that could be caused if the user "drifts" toward an incorrect pose, and is also useful in applications where gestures are used for continuous control.

To address these problems, we investigate *In Situ* ("in place" or "local") visual feedback (Figure 1) positioned directly in or on the hand. Rather than performing a gesture and waiting for discrete confirmation that it was recognized as intended, users see continuously updated, fine-grained feedback prior to terminating a gesture. Users can understand if one of their fingers is incorrectly positioned, and also learn to reproduce gestures more precisely. This helps users hit a smaller "target" in pose space, resulting in fewer recognition errors during subsequent use. This also means that there is more room in gesture space to later incorporate and learn additional gestures.

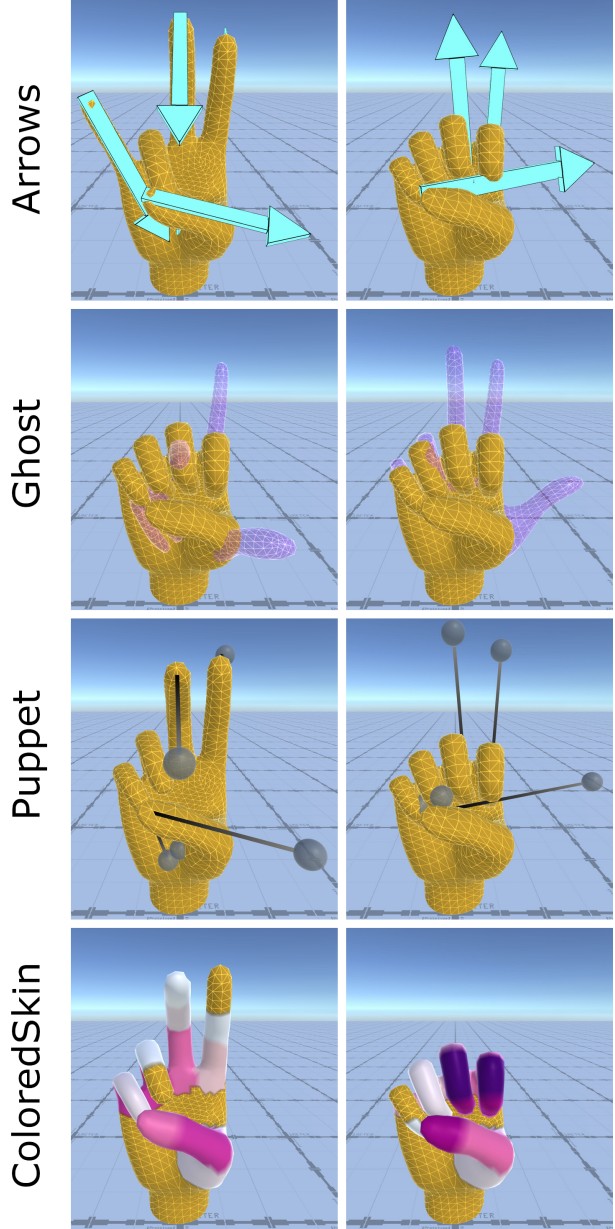

Figure 1: Four variants of In Situ feedback. The yellow mesh shows the user's current hand pose. Left column: the target gesture is with thumb and index extended. Right column: the target gesture is with thumb, index and middle fingers extended.

Our contributions are (1) the design of 4 variants of In Situ feedback, which, unlike previous work, provide in-the-hand visual aid for individual finger placement; (2) a discussion of design issues related to implementing such feedback; and (3) the results of two studies performed in virtual reality, collectively involving 32 users. Our 2nd study provides strong evidence that our novel Puppet variant

enables users to reproduce hand poses more precisely than a status quo cribsheet ("grid") and was preferred by most users.

## 2 BACKGROUND

Xia et al. [29] give a thorough, recent survey of issues related to gestural interaction. In our brief survey here, the term 'gesture' may refer to motions through space performed by a limb or input device, or refer to postures or poses adopted by a body part, or a combination of these. We review previous work on gestural interaction by considering different modalities for input, beginning with pointing devices in 2D or 3D space, and ending with the fingers of the user's hands.

### 2.1 Gestures with pointing devices

First, consider gestures made with the motion of a 2D pointing device such as a stylus, mouse, or a single finger on a touchpad. Typically, the user "draws" one of many possible gestures, or shapes, with the input device, causing a command to be selected. In some systems, the user must study the set of allowed shapes prior to using them, e.g., by using a "cribsheet", a visual key showing how each shape is drawn. A more advanced version of this is the GestureBar [7, 31], which allows users to look up and interactively rehearse gestures before using them in the workspace.

Other systems display visual guides *at the same time* that the user is drawing a gesture. This is sometimes called *feedforward*, in contrast to (confirmatory) feedback which is displayed after the system has recognized a gesture. (Throughout this paper, we use the term feedback to refer more generally to any visual indication, including feedforward as well as confirmatory feedback.) Feedforward can be displayed at the user's current location [1,5,6] to reduce division of attention and eye motion, or off to the side [1, 18] to reduce occlusion. Feedforward might be continually updated during the user's motion to only display the subset of gestures most closely matching the user's motion up until now, and/or only display the remaining portion of each possible gesture [1,5]. If only one gesture known to the system matches a user's partial gesture, the system can autocomplete the gesture [6], affording a shortcut to the user. Feedforward can also be displayed in the form of menu items, where the gesture's shape is implied by the location of the menu labels [4].

Although such feedforward makes it easier for beginners to perform gestures, it may also act as a crutch, making it less likely that users will remember the gestures. Anderson et al. [1] experimentally demonstrated that users remembered gestures better when the feedforward was hidden earlier and earlier prior to the user completing a gesture.

The preceding work has been for input in 2D, but OctoPocus [5] has notably been extended to 3D [9, 11].

### 2.2 Gestures with body parts

Next, we move on to gestures performed with the user's body, arms, or hands [14, 17], first considering gestures that do not require specific actions by individual fingers. Techniques for visually indicating how to perform a gesture with a body part include (1) showing a semi-transparent or "ghosted" image of the body part performing the gesture, which has been applied to the whole body [3, 30], the arms [10, 16], the hands [20] or an object held in the hand [27]; (2) showing a skeleton of the body part performing the gesture [2, 23]; or (3) using arrows [2,24]. We can classify these previous works into those that are *exocentric*, displaying indications in front of the user, e.g., on a screen, as if the user is looking at a mirror image or a 3rd person perspective [2, 23], and *egocentric*, with visual indications displayed in the 1st person on the user's own body in an immersive environment [10, 16, 20, 24, 27].

Next, there are gestural interfaces where the user's individual fingers must execute specific motions or poses. In one subset of this previous work, the gestures are performed on a multitouch surface. ShadowGuides [12] indicate how to perform multitouch gestures using silhouettes and arrows, and Arpège [13] displays labeled contact points for fingertips. We classify Arpège [13] as an example of *in situ* help, because the visual indications are displayed directly under the hand, in contrast to ShadowGuides [12] where the visual indications are *offset* to the side to avoid occlusion. (Note that Freeman et al. [12] describe their system as in-situ, but this is meant in a temporal sense, i.e., the visual help is provided at the same time that the user is executing gestures, to "learn while doing". We instead use the term "in situ" to refer to spatial coincidence.)

Another subset of work involves individual fingers moving or posing in 3D space. To visually indicate a gesture to the user, a common approach within VR games is to show the user exocentric static images or animations. GesturAR [26] displays exocentric animated skeletons of fingers. We are aware of no previous virtual reality interfaces employing in situ (on the bare hand) indications of how to move or position individual fingers. Our work studies this topic.

### 2.3 Other previous work

Our work was also informed by a previous study of hotkey learning by Grossman et al. [15], who experimentally compared different methods for learning shortcut keys in a pulldown menu. The condition that resulted in the fastest learning was one where users could *not* select menu items by clicking on them, and could only use the pulldown menu to look up the corresponding shortcut key. A pulldown menu that can only be used to lookup shortcut keys, and not click to select commands, is analogous to a cribsheet that can only be used to lookup gestures. If such a pulldown menu or cribsheet is inconvenient to access, this will incentivize the user to memorize the shortcuts or gestures, respectively.

## 3 TYPES OF GESTURES

This section considers hand gestures in a general sense, involving the hand either moving through space or positioning the fingers in a specific way. Table 1 identifies five types of such gestures. The term hand *pose* refers specifically to a configuration of fingers, i.e., a positioning of fingers in the hand's local space.

Table 1: Five types of gestures.

|  | Pose of fingers is not important | Pose of fingers is fixed | Pose of fingers changes over time |
|---|---|---|---|
| Hand does not move through space | (not applicable) | 2. Static Fingers | 4. Dynamic Fingers |
| Hand moves through space | 1. Dynamic Hand | 3. Dynamic Hand & Static Fingers | 5. Dynamic Hand & Fingers |

We consider each of the five types in turn.

1: *Dynamic Hand* gestures involve hand motions without depending on fingers. OctoPocus in 3D [9, 11] demonstrates one way to teach such gestures to users.

2: Gestures involving *Static Fingers* are invoked by adopting a particular pose. A simple example is the "pinch" gesture where the user touches the tips of their index finger and thumb. These are the kinds of gestures that we have designed for in our work. Compared to gestures with Dynamic Fingers, those with Static Fingers are more amenable to a clear definition of the start and end of a gesture, which can be useful for segmenting a stream of input.

3: Gestures with *Dynamic Hand & Static Fingers* are a combination of the previous two types. For example, pinching the thumb together with either the index or middle or ring finger could open one of three radial menus, after which the user moves their hand in one of four directions (north, south, east, west) to select an item in the radial menu, and then releases their pinch pose to complete the menu selection. The initiation and release of the pinch pose delimit the start and end of the gesture, but the pose remains fixed during the hand motion, hence we classify this gesture as having static fingers.

4 and 5: In the right-most column of the table, the pose of the hand changes over the course of the gesture. An example of this would be a pinch gesture where the user gradually spreads open or closes their thumb + index to zoom or resize an object. During this action, the hand could have a fixed position (*Dynamic Fingers*), or the hand could be simultaneously moving (*Dynamic Hand & Fingers*) to perform a translation of the scene or of an object. Additional examples in this column (in the *Dynamic Fingers* subset) are the "bloom" and "air tap" gestures on the Microsoft HoloLens, where the user must move between different poses (closed fist to extended fingers, or upward pointing index to downward pointing index, respectively) to invoke commands.

Our research on In Situ feedback is focused on the *Static Fingers* subset of the table, because there is a lack of previous work on visual aids for positioning individual fingers, and it makes sense to first design for static poses before considering dynamic fingers.

## 4 DESIGN OF IN SITU VISUAL FEEDBACK

### 4.1 The Importance of Precision

To reveal available gestures to new users, interfaces often use exocentric static images or recorded animations. A user attempting to reproduce such gestures may do so with small variations. For example, a hand pose where the index finger points forward might be done with varying placement of the thumb or smaller fingers. Such variations may lead to problems. First, if the gesture is not recognized as intended, the user may not understand what caused the error. Second, even if the recognition algorithm allows for such variations, this may encourage users to adopt imprecise poses that drift over time, resulting in subsequent recognition errors that are mysterious to the user. Third, allowing for more variation makes it difficult to later incorporate additional gestures that depend on correctly reproducing the precise placement of all fingers. Fourth, in applications that continuously interpolate between a set of hand poses (e.g., for continuous control of a virtual character's posture [19] or of a digital musical performance), it may be important for users to learn how to precisely reproduce key hand poses.

If we imagine pose space as a 2-dimensional keyboard, executing a gesture is like hitting a key. Without In Situ feedback, the user has no indication if they are hitting the center of a key, or near the edge. In Situ feedback should help users hit closer to the center of each key (i.e., perform gestures more precisely), allowing them to avoid errors later on, and also allowing the keys to be made smaller at a later time to introduce more keys within the same space.

### 4.2 Visual Design

Figure 1 shows variants of In Situ feedback which we implemented and evaluated. For clarity, Figure 1 shows snapshots of feedback from a fixed position in the hand's local space (i.e., the hand is rotated and centered at a fixed position).

Each type of feedback changes as the user's fingers move, updating to show the target gesture pose, except for Ghost feedback, in which the purple mesh remains fixed as the user's hand changes.

Note that each variant could be used in either virtual reality or augmented reality, so long as the user's hands are visible.

#### 4.2.1 Arrows

Arrows indicate where each of the fingertips should be positioned. To increase visibility, each arrow is rotated so that its flat side is facing the user as much as possible. Arrows that would be less than 3 mm long are hidden to reduce clutter.

#### 4.2.2 Ghost

Our Ghost feedback was inspired partly by the use of semi-transparent drawing of objects to suggest motion, seen in comic book art [22], and also in previous work such as Han et al. [16] and Dürr et al. [10] which show comparable feedback for the entire arm

rather than fingers. Unlike our other forms of feedback, the ghost feedback is static (not changing as the user's fingers move), which may be less distracting for the user as they seek to achieve the target pose. To prevent "z-fighting" (i.e., rendering artefacts caused by the user's hand and the In Situ feedback drawn with nearly coincident surfaces), the Ghost mesh is scaled down by 2% relative to the user's hand size.

#### 4.2.3 Puppet

In this feedback, target positions for fingertips are indicated with spheres which are connected via line segments to the user's current fingertip positions. Because the line segments resemble cords that are stretched taut, this feedback superficially resembles puppet strings.

The idea for this feedback came from redesigning Arrows to be as simple as possible. Tiny spheres are arguably the simplest way to indicate target positions. To show the association between these spheres and each finger, the simplest approach is arguably straight line segments. The result creates less occlusion of the hand.

#### 4.2.4 ColoredSkin

ColoredSkin paints a heatmap on the skin, showing the difference between the user's hand and the target gesture with a color gradient. Each joint is colored darker if a larger rotation is needed to achieve the target gesture. Unlike the other types of feedback, ColoredSkin does not directly indicate the target position of each finger, which may make it more difficult to understand. However, the increased effort required by a user to achieve a target gesture may work in favor of remembering the gesture, as suggested by previous studies involving effort and memory [1, 8, 15].

### 4.3 Uses of In Situ Feedback

One way to use In Situ feedback is as part of a tutorial mode where the system is teaching the user one or several gestures. Each time the system prompts the user to perform a gesture, the system can display the corresponding In Situ feedback. We call this **system-driven** feedback. This is analogous to a system asking the user to invoke a search command by hitting a shortcut key combination like Ctrl+F or ⌘+F. However, users may grow impatient with such tutorials if there are many gestures to learn.

Continuing with the analogy, keyboard shortcuts are often listed inside a pull-down menu where the user can simply click on the name of a command. Users are often not willing to invest time learning many shortcuts in a tutorial and might prefer to just open the pull-down menu as needed. Once the pull-down menu is open, there is a natural tendency to simply click on the desired command without reading the associated keyboard shortcut. Grossman et al. [15] evaluated alternative interfaces where the user could open a pull-down menu to find a command, but then *had* to hit the corresponding keyboard shortcut, helping the user to memorize the shortcut. A similar approach is possible with In Situ feedback: the user would first select the desired command in some kind of menu, causing the corresponding In Situ feedback to appear, and the user would then be required to perform the gesture, helping them memorize that gesture for faster subsequent invocation. We call this **user-driven** feedback. This is not a tutorial in the sense of the previous paragraph, since now the user is selecting the desired command each time, perhaps as part of real work tasks. Once the user has learned a gesture, they can perform it without first opening the menu.

A third possible use is for the system to detect whenever the user's hand pose is close to a gesture $G$ known to the system, at which point the system displays the In Situ feedback for $G$. We call this predictive feedback an **autocompletion** hint. This is analogous to the user holding down the Ctrl key, at which point the system displays "Ctrl+F: Find" because that is the only (or the most common) shorcut key combination that starts with Ctrl.

To implement autocompletion hinting, our code used a temporal threshold $\tau_t$ and distance threshold $\tau_d$: if the user's pose remains within distance $\tau_d$ of a known gesture for time $\tau_t$, this triggers the displaying of In Situ feedback. Our method for calculating distance is explained in Section 5.1. We set $\tau_d = 12.6$ cm for both our studies. In Study 1, $\tau_t = 1000$ ms, and in Study 2, $\tau_t = 500$ ms.

## 4.4 How to Terminate a Gesture

Another design issue is how to allow a user to terminate (i.e., confirm) a hand pose gesture. This may be difficult to do with the same hand and without changing the gesture recognized by the system. Possible strategies include: once a gesture is executed, (1) the user might hold their hand in the appropriate pose and await a timeout (where the progress of the timeout is shown with a timer ring or similar feedback), however this could be tiring for the user and also make the interaction feel less "fluid"; (2) the hand performing the gesture could perform a sudden lateral movement to signal completion, however this risks recognition errors depending on the tuning of the system; (3) the *other* hand could press a hardware button (such as a wireless clicker), or press a software widget, or perform a "completion" gesture such as touching the index and thumb (of the other hand) together – this requires the 2nd hand but avoids the problems of the other strategies just mentioned.

## 5 Implementation

Our system was developed with Unity and a Meta Quest 2 headset.

As part of our research, we implemented software infrastructure allowing us to record, playback, process, and visualize sequences of hand poses. This allowed us to test, debug, and fine tune features such as gesture recognition, autocompletion, and the visual design of feedback without wearing a headset, greatly accelerating the development process.

### 5.1 Defining the Distance between Hand Poses

To recognize a gesture, the system must find which $p_i$ of the known poses $\{p_i\}$ is most similar to the user's current pose $p_u$. Doing this requires defining the distance between any two poses. This distance is computed by aligning the two poses at the wrist, and then quantifying the dissimilarity between $p_i$ and $p_u$. We experimented with various formulations of this dissimilarity, taking into account the positions of fingertips, or the positions of all joints in the hand, or the distances between joints and the wrist, and we compared these alternative formulations by testing how they ranked the similarity of a pool of known poses to a variety of input poses. We found that the simplest formulation, based on fingertip positions, produced rankings that were subjectively no worse than other formulations. In our fingertip formulation, the distance between poses $a$ and $b$ is $\Sigma_{k=1}^{5}\|f_{k,a} - f_{k,b}\|$, where $f_{k,p}$ is the 3D position of the $k$th fingertip of pose $p$. Notice that this distance has physical units, namely the sum of distances between corresponding fingertips, which we measure in centimeters. Because this distance function only uses the 5 fingertips, it is faster to compute than a distance based on all joints, and is more portable between different hand-tracking platforms for the following reasons: some platforms might not track all internal joints, or not track them reliably, and different hand tracking platforms may decompose the hand into different sets of internal joints. The fingertip positions are a lowest common denominator that are more likely to be reliably tracked across platforms.

Our distance function weighs all fingers equally, however future work could evaluate if different fingertips should be given different weights.

## 6 Study 1

Study 1 was designed to answer two research questions: first, does In Situ feedback help users to be more precise in performing gestures

than status quo feedback, and second, which form of In Situ feedback is most effective?

### 6.1 Conditions

In a status quo VR user interface with hand gestures, the user might have a way of accessing a visual guide (sometimes called a "cribsheet") to learn the available hand gestures. This guide might be displayed off to the side, requiring the user to rotate their head, or might be popped open like a menu whenever the user presses a button. We implemented this feature and called it the "grid" (Figure 2) because it can display multiple rows and columns of gestures, each with a corresponding command name, somewhat like a display case. The hands in the grid are 3D models with depth. Our grid is displayed directly in front of the user, and only displayed when the user holds down a button. This allows us to measure how often the user accesses the grid and also allows us to impose a cost to access the grid in the form of an animated opening of the grid, to incentivize memorization. (Grossman et al.'s previous study [15] of hotkeys imposed an analogous cost, in that the user had to move their cursor from the bottom to the top of the screen before opening a pull-down menu, unless they had memorized the shortcut key.) A pilot study convinced us of the need to impose such an inconvenience to motivate users to learn gestures, otherwise users tended to simply open the grid repeatedly without trying to memorize them. In our Study 1, the duration of the grid's opening-up animation was 2500ms.

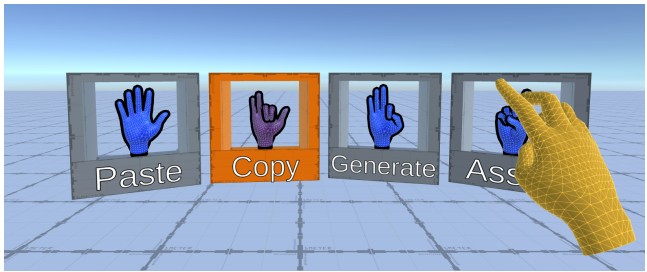

Figure 2: The *grid* is a kind of help menu or cribsheet that the user pops open, and is extensible to multiple rows and columns. In both our studies, the grid showed 1×4 gestures.

The first condition in our experiment is a *Baseline* condition which only gives the user access to the grid, without any In Situ feedback on their hand. In this Baseline condition, the user simply opened the grid to learn the required gestures and perform them.

There was also an additional condition for each of the variants of In Situ feedback (Figure 1). In these conditions, when the user opened the grid, they could select a gesture and it would activate a **user-driven** In Situ feedback on the user's hand (as described in Section 4.3) to guide them toward the correct gesture. The user could also trigger **autocompletion** In Situ feedback (whether or not they had opened the grid), simply by maintaining a hand pose close enough to a known gesture for enough time.

In the Baseline condition, the grid could be opened but the user could not select within it, because there was no In Situ feedback to display.

In all conditions, the user was not obliged to open the grid if they had already memorized the gesture.

### 6.2 Task

Users performed gestures with their right hand while holding a wireless clicker in their left hand (Figure 3). Each trial was initiated by the user pressing a button #1 on the clicker to signal that they were ready to begin.

The system would then display the name of the command whose corresponding gesture had to be performed. If the user had not yet

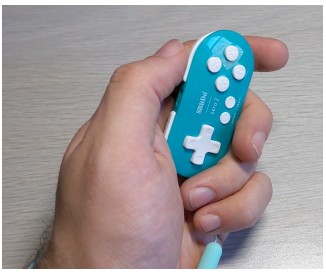

Figure 3: The wireless clicker held in the user's left hand.

memorized the corresponding gesture, they used a button #2 on the clicker to pop open the grid, which rose up from the ground plane with a 2500 ms animation.

Releasing button #2 dismissed the grid, causing it to disappear. The user could perform the gesture with their right hand, either before or after dismissing the grid. (However, it was only after the grid was dismissed that In Situ feedback was displayed.)

Once they were satisfied with their gesture and button #2 was released, they had to press button #1 to terminate the gesture and end the trial.

This caused a ring timer to appear that would gradually fill itself (Figure 4) before the next trial began. To motivate users to perform gestures with greater precision, the duration of the ring-filling animation was longer if the user's gesture had been less precise. The delay imposed by the ring timer also forced users to rest their hands before the next trial. There was no need to maintain any hand pose during the ring-filling animation.

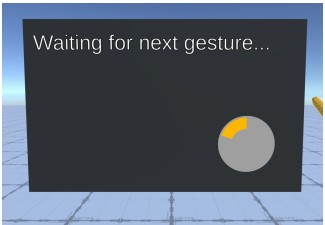

Figure 4: At the end of each trial, a ring timer (in orange) was displayed and gradually filled, forcing the user to pause before the next trial. The ring timer filled more slowly if the user was less precise in the gesture they had just performed.

In all conditions except for Baseline, the user could select a gesture within the grid, causing In Situ feedback to appear on their right hand. Selection within the grid was done by holding down button #2 to keep the grid open, and using the head direction to select a gesture in the grid. The frame around the gesture would highlight in orange in response to the head direction, and after a timeout of 500 ms, the hand showing the gesture would also change color, to indicate that the gesture was selected. This timeout was to prevent accidental misselections during rapid sideways head motion when releasing the button, as users tended to look at their hand just before releasing.

The precision of the gesture performed by the user was quantified with a distance $d$ in centimeters to the target pose (see Section 5.1). A pilot study found that $d$ varied between 1 and 35 cm, and these were mapped to a duration for the ring-filling animation using a ramp function varying from 1 to 10 seconds. Specifically, the duration in seconds was $\min(\max(((10-1)/(35-1))(d-1)+1,1),10)$.

To recap, user performance was incentivized in two ways: first, having to wait 2500 ms each time the grid was opened discouraged users from opening it too many times and encouraged memorizing

the gestures, and second, having the ring-filling animation's duration increase with fingertip distance encouraged users to be more precise with their gestures.

The random association of command names to gestures was done once for each user, such that a user never saw the same gesture associated to different command names or the same command name associated to different gestures, to avoid confusion.

### 6.3 Gesture Set

The set of gestures for study was generated in two steps. First, $2^5 = 32$ canonical hand poses were generated covering every combination of each finger being extended or closed. Second, to sample more of pose space, informal data collection was done, asking two pilot participants to perform spontaneous, widely varying poses with their hands for 2 minutes each, while the system recorded their hand. We algorithmically sampled these recordings, choosing random poses that were not too similar to already-collected poses, until we had a set of 101 gestures in total.

Of these 101 gestures, 2 with the middle finger extended were eliminated as they are considered obscene and users felt uncomfortable performing them. The remaining 99 gestures (Figure 5) were used in Study 1. The trials for the conditions (Section 6.1) and main task (Section 6.2) were generated for each user by sampling from this set of 99.

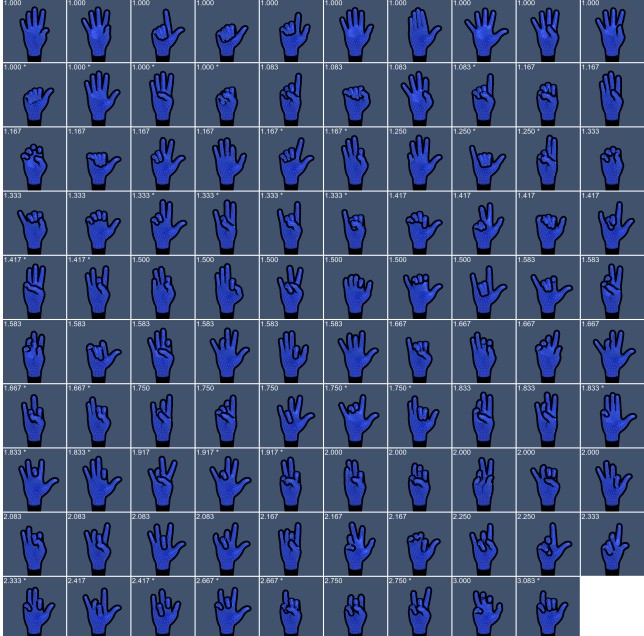

Figure 5: The pool of 99 gestures used in Study 1, sorted by average difficulty score (shown in upper left corners) as rated subjectively by the 12 participants. Scores followed by a star indicate canonical gestures.

After the trials were complete, each of the 12 users in Study 1 was also asked to rate the difficulty of each of the 99 gestures, on a scale of 1 (easy), 2 (medium), 3 (difficult), and 4 (impossible). These ratings were used to reduce the set of gestures used in Study 2, as described later.

### 6.4 Protocol

Equipment was disinfected before and after each user session. At the start of each session, after signing a consent form, users filled out a pre-questionnaire and had their interpupillary distance (IPD) measured, and the headset was adjusted for comfort and for the

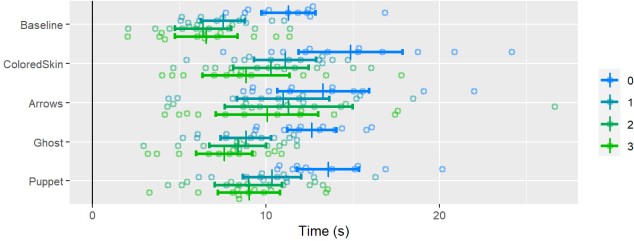

Figure 6: Duration of trials in Study 1. As users advanced through the blocks of each condition, their time tended to decrease. In this and all subsequent charts, colors show block number, each dot is for one user, and error bars show 95% confidence intervals.

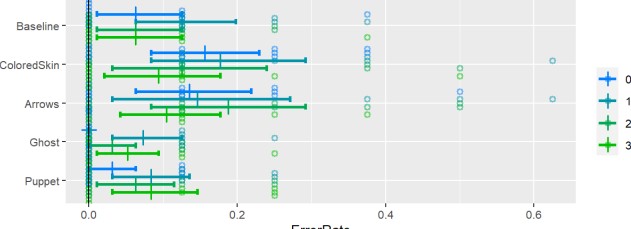

Figure 7: Error rates in Study 1. In a given trial, if the user's performed hand pose more closely resembled an incorrect gesture than the target gesture, this was counted as an error.

measured IPD. Users were then shown printouts to explain how to proceed in trials of all the conditions of the experiment. After the trials were completed, and each of the 99 gestures had been evaluated, a post-questionnaire was filled out.

### 6.5 Users

12 users were recruited: 9 men, 3 women; 10 right handed, 1 left handed, 1 ambidextrous, but all with a habit of using the mouse with their right hand; age 20 to 46 years (average 27.4); IPD 54 to 65mm (average 61.0), all with previous experience using 3D software and headsets. Each user was asked if they engaged in activities requiring well developed manual dexterity such as playing a musical instrument, drawing, or soldering; 10 out of 12 answered yes.

### 6.6 Design

Each user experienced the 5 levels of the Condition variable {Baseline, Arrows, Ghost, Puppet, ColoredSkin} in random order. For each Condition, the user performed a single warmup block followed by a sequence of 4 real blocks.

For the warmup block, the grid was populated with 2 gestures selected at random from the pool of 99 gestures, and 4 trials (in random order) were generated asking the user to select each gesture twice (i.e., 2 gestures × 2 repetitions, in random order, yielding 4 trials).

For the sequence of 4 real blocks, the grid was populated with 4 gestures selected at random from the pool of 99, and 8 trials (in random order) were generated for each block, asking the user to select each gesture twice (i.e., 4 gestures × 2 repetitions, in random order, yielding 8 trials per real block).

Gestures were chosen so that the same gesture never appeared in different conditions or different sequences for the same user.

There were a total of 12 users × 5 levels of Condition (in random order) × 4 real blocks × 4 gestures × 2 repetitions = 1920 trials, not counting warmup trials. Each user session lasted ≈ 90 minutes, of which ≈ 50 minutes were spent in VR.

### 6.7 Results

Figures 6-8 show the average time, error rate, and precision (quantified as distance) for Study 1. The Baseline condition yielded some of the lowest times, but also some of the worst distances, which is not surprising since users had no In Situ feedback to attend to. In terms of distance, the best conditions were Ghost and Puppet, providing evidence that In Situ feedback can indeed help users to be more precise. Ghost and Puppet are also better than the other two In Situ conditions with respect to error rate.

Figure 9 shows the subjective ratings of the users for all conditions. In Study 1, Ghost and Puppet were preferred by most users, and both conditions resulted in favorable Likert scores, with Puppet obtaining the best Likert scores for two criteria (Physical effort and

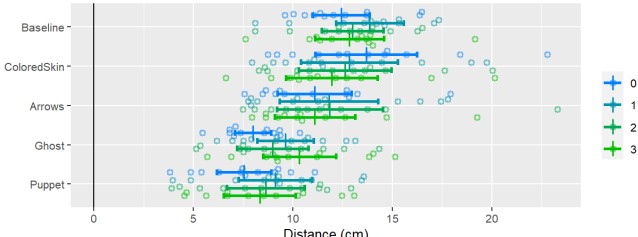

Figure 8: The "distance" (sum of distances between fingertips) between the user's performed hand pose and the target gesture, in Study 1. This quantifies the precision of the user's gesture. Puppet was one of the most precise conditions.

Frustration), tying for a third (Enabled task), and 2nd best for Mental effort.

In Study 1, because the threshold for autocompletion hinting was $\tau_t = 1000$ ms, this gave users some freedom to deliberately wait for it to trigger without first opening the grid, or conversely, to open the grid and select within it (to invoke user-driven In Situ feedback) and terminate the gesture before autocompletion hinting was triggered. We asked users if they preferred autocompletion, and 10 out of 12 answered 'yes'.

### 6.8 Discussion

We were surprised that Arrows did not do better in terms of error rate and distance, because Arrows seem like an obvious kind of feedback to display. Subjectively, however, the Arrows can be distracting because they continually change size and orientation as the user's hand pose changes, whereas the Ghost and the spheres of the Puppet remain static with respect to the user's hand and are perhaps easier to perceive and remember as a single rigid object. Puppet also results in less occlusion than Arrows because the line segments are thinner than arrows, and Puppet feedback is perhaps also less distracting because strings stretched like elastics conform to a physical metaphor, contrary to Arrows that continually change size as their length changes.

It is also possible that Puppet has an advantage over Ghost be-

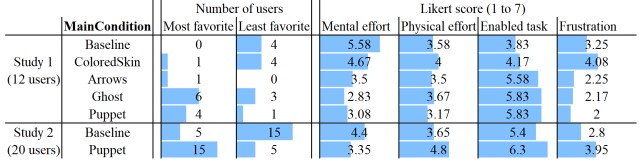

| | | Number of users | | Likert score (1 to 7) | | | |
|---|---|---|---|---|---|---|---|
| | **MainCondition** | Most favorite | Least favorite | Mental effort | Physical effort | Enabled task | Frustration |
| | Baseline | 0 | 4 | 5.58 | 3.58 | 3.83 | 3.25 |
| Study 1 | ColoredSkin | 1 | 4 | 4.67 | 4 | 4.17 | 4.08 |
| (12 users) | Arrows | 1 | 0 | 3.5 | 3.5 | 5.58 | 2.25 |
| | Ghost | 6 | 3 | 2.83 | 3.67 | 5.83 | 2.17 |
| | Puppet | 4 | 1 | 3.08 | 3.17 | 5.83 | 2 |
| Study 2 | Baseline | 5 | 15 | 4.4 | 3.65 | 5.4 | 2.8 |
| (20 users) | Puppet | 15 | 5 | 3.35 | 4.8 | 6.3 | 3.95 |

Figure 9: Subjective ratings by users for Studies 1 and 2, shown together to ease comparison.

cause the distance between gestures is measured between fingertips, whose ideal positions are indicated directly with Puppet, whereas Ghost shows a slightly smaller version of the ideal hand pose.

Users commented that some conditions seemed more difficult than others because the hand poses were more difficult to achieve. The assignment of gestures to conditions was random and different for each user, however we realized that a subsequent study could increase statistical power by better controlling hand pose difficulty across conditions. We therefore planned a 2nd study to collect more data on the most promosing In Situ feedback, namely Puppet, that would assign gestures of roughly equal difficulty to each condition.

## 7   STUDY 2

The results of the previous study suggest that the main advantage of Puppet is in the precision of the gestures performed by the user, as measured by fingertip distance (Figure 8). Thus, the main **hypothesis** we test in Study 2 is that Puppet results in smaller distances compared to the Baseline condition. To increase statistical power in Study 2, we recruit more users who will perform more trials in Study 2, and we only compare the two conditions of Puppet and Baseline. Section 7.3 explains how we controlled the difficulty of gestures in each condition to have less variability than in Study 1.

We also want to better understand how much autocompletion can help users. We reduced the trigger time from $\tau_t = 1000$ ms to 500ms so that autocompletion would be suggested earlier. Autocompletion was also modified to display the predicted command name on the back and front of the user's wrist (Figure 10). We also modified our software to measure more data about how the grid and autocompletion was being used during trials.

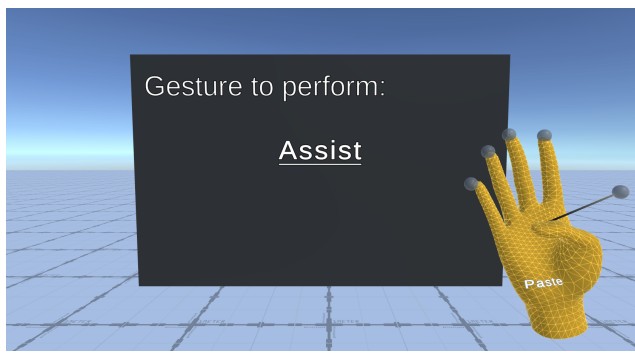

Figure 10: In Study 2, in the Puppet condition, the name of the predicted command for autocompletion was displayed on the wrist. In this case, the system is asking the user to perform the gesture for Assist, but the user's hand pose is currently closer to the gesture for Paste, which is being suggested by the Puppet feedback and displayed on the wrist. The user may correct their pose before terminating the trial.

To further encourage users to memorize gestures and not overly rely on the grid, the animation of the grid opening was modified to become gradually slower during each trial. In Study 1, the grid always took 2500ms to open. In Study 2, the time varied as a function of the block and also the number of times the grid had been invoked within a trial. Specifically, to open the grid the $n$th time in a trial of the $b$th block, where $n \geq 0$ and $b \geq 0$, the animation lasted $(2500 + (b - 1 + n - 1) \times 300)$ ms.

### 7.1   Protocol

The protocol for Study 2 was similar to that of Study 1, with two changes. First, to provide a more complete explanation of trials before starting, users were asked to hold the clicker, without the headset, to try the buttons and have each button's function explained.

A printed document showed how to do gestures with puppet feedback, and users were asked to complete the gestures in the printed document before beginning the experiment. Second, users did not rate the difficulty of gestures at the end of the session.

### 7.2   Users

20 users were recruited: 10 men, 10 women; all right handed; age 21 to 39 years (average 22.6); IPD 55 to 70mm (average 59.9); 14 out of 20 with previous experience using 3D software; 13 out of 20 with previous experience using headsets.

Each user was asked if they engaged in activities requiring well developed manual dexterity, such as playing a musical instrument or drawing. 8 out of 20 answered yes.

### 7.3   Design

Each user experienced 2 levels of Condition {Baseline, Puppet} in random order. For each Condition, the user performed two warmup blocks followed by two sequences of 4 real blocks. For each warmup block, the grid was populated with 2 gestures selected at random, and 4 trials (in random order) were generated asking the user to select each gesture twice (i.e., 2 gestures × 2 repetitions, in random order, yielding 4 trials in each warmup block). For each sequence of 4 real blocks, the grid was populated with 4 gestures selected at random, and 8 trials (in random order) were generated for each block, asking the user to select each gesture twice (i.e., 4 gestures × 2 repetitions, in random order, yielding 8 trials per real block). There were a total of 20 users × 2 levels of Condition (in random order) × 2 sequences × 4 real blocks × 4 gestures × 2 repetitions = 2560 trials, not counting warmup trials. Each user session lasted ≈ 75 minutes, of which ≈ 35 minutes were spent in VR.

In Study 1, each subset of 4 gestures for each sequence of blocks was simply chosen at random from the pool of 99 gestures. This meant that the difficulty of getures was quite variable from one sequence of blocks to another. In Study 2, to reduce this variability, we chose subsets of 4 gestures that satisfied some constraints which we now describe.

Recall that Study 1 asked users to rate the difficulty of each of the 99 gestures, from which we computed the average difficulty of each gesture. For Study 2, from the original pool of 99 gestures, we chose 12 "easy" gestures (with difficulty between 1.0 and 1.33) and 12 "medium" gestures (difficulty between 1.5 and 1.92), chosen so that none appear too similar to each other. This produced a new pool of 24 gestures. The first constraint on each subset of 4 gestures is that they were chosen from this reduced pool of 24. The second constraint is that the total difficulty of the 4 gestures must be within 5% of 4 times the average difficulty of all the gestures. The third constraint is that we required the distance (as defined in Section 5.1) between any 2 gestures in the subset of 4 be at least 15cm. Finally, gestures were chosen so that the same gesture never appeared in different conditions or different sequences for the same user (i.e., for each user, 8 of the 24 gestures were used for warmup blocks, and the other 16 for real blocks).

### 7.4   Results

Figures 11-13 show the average time, error rate, and distance, per condition and per block. Figure 14 shows distances aggregated across blocks, along with paired differences calculated as part of a $t$-test providing strong evidence that Puppet results in smaller distances (i.e., greater precision) than the Baseline, confirming our main hypothesis.

Figures 15 and 16 show the average number of times the grid was opened, and the total time the grid was opened, by condition and block. Decreasing trends within each condition are apparent.

We also found that in more than 90% of the Puppet trials, autocompletion was triggered and displayed the correct gesture. This rate was roughly constant across blocks with no obvious trend.

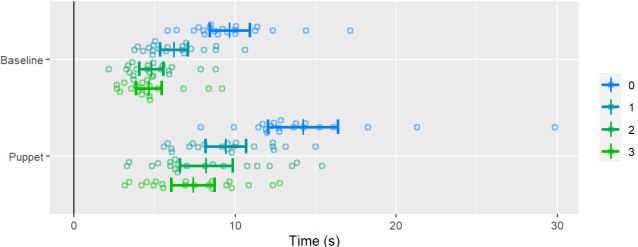

Figure 11: The duration of trials in Study 2. Performance improved as users advanced throught the blocks.

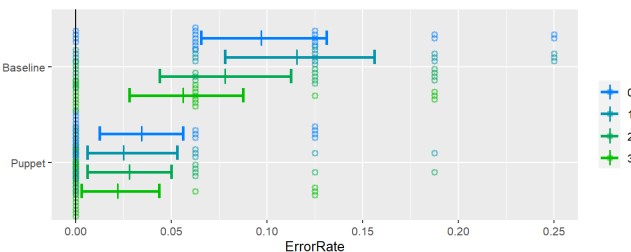

Figure 12: Error rates in Study 2.

Although the autocompletion hint was triggered when the user was within $\tau_d = 12.6$ cm of a known gesture for the minimum time of $\tau_t = 500$ ms, our recorded data indicates that users usually got closer to the target gesture before the autocompletion hint was triggered (Figure 17).

Figure 18 shows that users often spent 3 seconds or more correcting their hand pose after the autocompletion was triggered, even during the last block, presumably to improve the precision of their gesture and reduce the penalty time imposed by the ring timer.

We also calculated the average duration and average distance for each user within the Puppet condition, resulting in 20 pairs of numbers, and checked for a correlation between these two variables. The Pearson correlation coefficient was $-0.52$ with $p < 0.05$. In other words, users who spent more time performing a gesture resulted in lower distances.

Figure 9 shows that Puppet was preferred over Baseline by 15 out of the 20 users, and scored better on Mental effort and Enabling the task, but worse on Physical effort and Frustration. These scores may reflect the fact that Pupper made incorrect poses visible to the user, pressuring them to correct their pose, whereas Baseline did not.

We checked but found no evidence of differences in time, error rate, or distances between the two gender groups, nor between the group of users who reported activities involving more dexterity

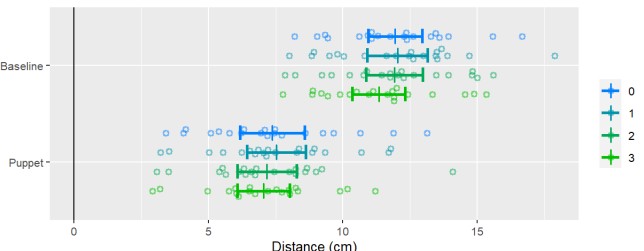

Figure 13: The distance (sum of distances between fingertips) between the user's performed hand pose and the target hand pose, in Study 2.

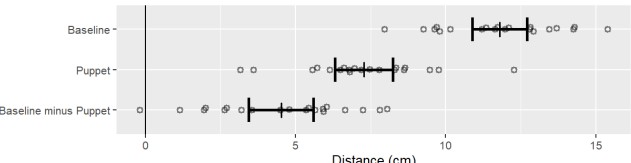

Figure 14: Distances aggregated across blocks, in Study 2. The 3rd error bar shows the result of a paired t-test. Because the zero line falls far outside this 3rd error bar, the t-test yields a significant result, with $p < 5 \times 10^{-8}$. Puppet is therefore more precise.

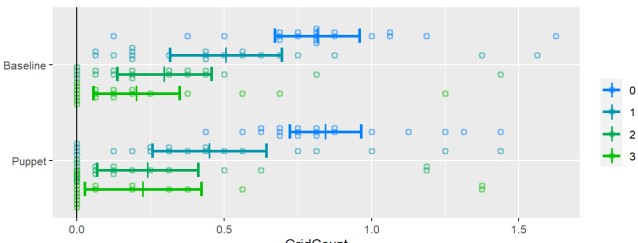

Figure 15: The number of times the grid was opened during a trial, during Study 2.

versus those who did not.

### 7.5 Discussion

The decreasing trend seen in Figures 15 and 16 indicate that users were relying less and less on the grid as they advanced through blocks, presumably because they were memorizing the gestures (at least approximately) and using this memory to reproduce the gestures, often with help from autocompletion.

The inverse correlation between distance and duration suggests a speed-accuracy tradeoff, which is also found in other interaction tasks [21].

### 8 CONCLUSIONS

We have presented the design and experimental evaluation of In Situ visual feedback, the first egocentric visual feedback for positioning individual fingers to learn hand gestures. Of the four variants we implemented, Puppet feedback resulted in the most precisely reproduced gestures, as measured by distances between fingertips (Figures 13 and 14), and was also preferred by most users (Figure 9).

### 9 FUTURE DIRECTIONS

In Situ feedback might be applied for rehabilitation of patients after injury or surgery to their hand, or for the training of musicians.

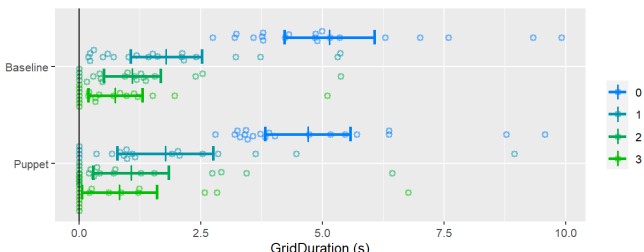

Figure 16: The total time spent with the grid opened during a trial, during Study 2.

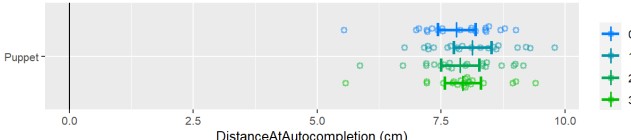

Figure 17: The distance between the user's hand pose and a known gesture at the instant that autocompletion was triggered.

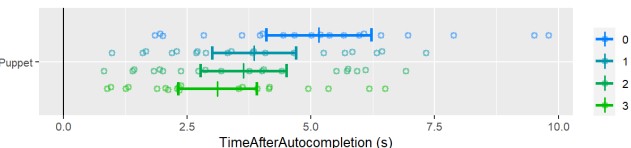

Figure 18: The time it took after autocompletion was triggered for the user to terminate the trial.

Our autocompletion hint is limited to only displaying one predicted gesture at a time. It may sometimes be advantageous to display multiple predicted gestures, for example the *N* known gestures most similar to the user's current hand pose, near the user's hand, possibly in a radial layout around the hand.

Future work could also investigate In Situ feedback for the other kinds of gestures in Table 1.

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
