# OpenReview forum: "In Situ Visual Feedback for Learning Hand Gestures in VR"
_graphicsinterface.org/Graphics_Interface/2023/Conference — Submitted to GI 2023_

### Official Review · Reviewer_1Z3G · 2023-01-05
**Interesting topic but in need of polish and clarity**

**Rating:** 4
**Confidence:** 5

**Review:**

This submission presents four “in situ” feedback types that can be used within VR to teach users the propose hand poses to use. It also presents the results of two studies. The first evaluated the four feedback types, in addition to a baseline condition, along the dimensions of duration, distance, and error rate and found that the baseline / ghost / were performed fastest, baseline, ghost, and puppet had the smallest error rates, and puppet and ghost had the smallest distance. A second study compared puppet to the baseline along the dimensions of duration, error rate, distance, # of times assistance was used, and duration the assistance was used for. The results seem to suggest that participants were faster with the baseline, had less error with puppet, had a smaller distance with puppet, opened the assistance approximately the same amount, and spent similar amounts of time using both techniques. The results from this submission, and the four techniques themselves, form the contribution and are relevant within the scope of GI.

So this paper presents some interesting feedback types and I appreciated the diversity of the ideas. Enhancing how we learn gestures and ensuring that we can perform them so as to not cause errors (human or system) is an important area of research. With this being said, I don’t quite know if I agree that this paper is looking at gestures, hand poses yes (and the submission uses the term hand poses quite a bit), but the vagueness of the definition – which includes motion through space and static poses seems too wide of a scope. This becomes a bit problematic when looking at Figure 1, where things like the arrows or the ball and lines suggests that the “gesture” is composed of motion rather than telling the user where to move their fingers to make a static pose - why not just show them the static pose overlain on their hand with different transparency levels? Something just feels off and confusing about this gesture / hand pose / lets show you where your fingertips should be. The examples in Figure 1 need to be improved to show the target gesture in one column, the feedback in the other? Maybe this will help clear things up? I wasn’t understanding what the user should be doing based on the images alone. I probably would have scoped this paper to focus on hand poses only.

The text in the intro also needs to be more clear that the paper focuses on in situ (i.e., egocentric feedback) rather than simply feedback in the same environment. There is a Note in 2.2 about this, but the note needs to be made earlier to provide clarity.

While I appreciated what Section 3 (Types of Gestures) was going for, I found this table and explanation to be unclear. I was uncertain why the fingers were divided into static and dynamic but the hand was not. I was also confused as to why the text first presented the dynamic hand, then switched both dimensions to static fingers, then combined the two before explaining what would be meant by dynamic fingers, and then combined 4 and 5. I found this very confusing and I am still unclear what the difference between the two pinch gestures from 2 and 4 is. I would recommend first describing 2, then 4, then 3, then 5, and then talking about the special case of 1.

In terms of the feedback types themselves, it is unclear why these four ideas were implemented over, say just placing colored dots (one color per finger) or different shapes (one shape per finger) at the correct locations in space (4.2.3 alluded that dots were the simplest way to indicate target positions). It was also unclear why 4.2.4 was chosen (no reference to prior work or inspiration) over say, a semitransparent rendering of the users hand with the target gesture hand overlayed. Lastly, it was unclear why ghost, which was inspired by motion, would be a suitable idea for static hand poses.

For study 1, it was unclear why it was important to evaluate (almost all) finger extensions / contractions, when many of these hand postures are very unnatural and would likely never be implemented within a working system (they also don’t map naturally to existing metaphors or functionality – e.g., the first two gestures that had a 1.917 difficulty, gesture with the difficulty of 3.000, etc.). More justification for why this variety of hand poses was used is needed. In the study 1 results, it is unclear why statistics were not run on the data, how the error rate was computed (i.e., “how was more closely resembled” measured), how error rate differed from distance, why the puppet error rates went up from block 0 to 4, why the distances for ghost (and somewhat baseline, arrows, and puppet) showed an upward trend from 0-4, and what the anchors on the Likert scale was. For the Likert data, because Likert data is ordinal, medians and an error measure such as interquartile range should have been used. Note that providing means without error measures is a dangerous practice.  Lastly, were participants asked to indicate why they liked or didn’t like specific feedback types? If there are any participant quotes it would be useful to include them.

For study 2, justification is needed for the four constraints placed on the gesture pool. Why 5% of 4 times the average? Why a distance of 15 cm? These seem arbitrary. Test statistics are also missing for the paired t-test, along with results of the other paired t-tests (that I’m going to assume were also run – or if not, why weren’t they run for the other measures?). I also didn’t quite understand this statement given that there doesn’t appear to be much of a change in error rates or distance across the blocks: “The decreasing trend seen in Figures 15 and 16 indicate that users were relying less and less on the grid as they advanced through blocks, presumably because they were memorizing the gestures (at least approximately) and using this memory”.

In conclusion, while this submission is interesting and I appreciate how much information was fit into the small number of pages, there are too many unanswered questions and confusing elements that have discouraged me from recommending it for acceptance in its current state.

Other:
-	How is the research mentioned in 2.1 relevant? A tieback to the present exploration is needed.
-	1st and 2nd should be written out as First and Second.
-	Don’t start a sentence with a number – spell out the word instead (e.g., 20 vs Twenty).
-	A transition sentence / paragraph is needed between 4 and 4.1 to intro what 4 contains
-	The paper makes repeated reference to [15], however this seems out of place, especially in 4.3, which the second paragraph could be shortened substantially to explain user-driven feedback  (which isn’t a complicated idea to be honest)
-	Each block should start at 1 rather than 0, especially as there were 4 “real blocks” and one warmup block (i.e., warmup = 0, and 1-4 are real)
-	By combing Study 1 and 2 in Figure 9, this naturally encourages a comparison, however, because the experimental conditions were different, this should be split into two tables
-	The table placement needs work to improve readability – place tables as close to the text that describes them as possible. Why are 6 and 7 in the same row but then 8 is a row down but on the right column? This makes the paper difficult to read.
-	This phrase is unclear as no statistics were run in study 1: “however we realized that a subsequent study could increase statistical power by better controlling hand pose difficulty across conditions.”
-	“users were asked to complete the gestures in the printed document before beginning the experiment” – how did participants “complete the gestures”? this is unclear
-	Study 1 should include details on previous experience with 3d software and headsets like study 2 does
-	Were any of the participants paid? How were they recruited?
-	If the metric is named average time, use the same term throughout the text (i.e., don’t switch to duration)

---

### Official Review · Reviewer_kYGj · 2023-01-05
**Interesting area of work, but necessary detail and depth missing.**

**Rating:** 4
**Confidence:** 4

**Review:**

This paper presents an exploration of five different types of gesture guides, and evaluates how well people could perform them as measured by error (distance to correct pose), time, and menu invocations. Overall this paper explores an interesting direction as AR and VR headsets become more prevalent and users need more access to rapid functionality that is not located behind a GUI that occludes their screen.

Overall, I don't believe this paper is ready for publication at GI 2023. I believe there are some significant details missing, and the analysis is not deep enough to form a strong takeaway from the paper. I also think the paper needs to explore the tradeoff between learning and performance - in its current form, the paper seems to conflate these a bit despite citing some earlier work highlighting the difference, and one sentence in the discussion.

I did appreciate the two studies. The initial study did a good job of testing some varied design concepts and setting up the second study which had more rigor. I also thought the figures were clear and appreciated seeing all of the data as well as error bars. The error bars should have clear labelling in the caption though (are they SD? SEM? Confidence intervals?).

In general though, I felt the paper lacking for details, particularly around the experiments result and analysis, and some of the study protocol details.

What does it mean to 'algorithmically sample' the recordings?

In Study 2, how were the difficulty thresholds decided?

For the t-tests, the values (e.g., test statistic) should be given, and there should also be completeness for testing (i.e., all tests that were run should be fully reported). Right now, it's not clear what was compared, vs not run. For instance, I don't know what t-test is being reported in the caption of Figure 14 - is that baseline vs. Puppet, or is it referring to something with the 'baseline minus puppet' (which I'm confused as to why that was included).

More details on the implementation of pose similarity would be useful. The statement that the final method (distance of 5 fingertips) was subjectively no worse than other formulations should be supported in greater depth. What other formulations were tested? How were they evaluated? It would also be good to clarify whether or not the pose similarity method is considered a contribution of the paper - it is simple and fast to compute, so if validated properly it might serve as its own contribution.

Some of these issues can be fixed in a revision, but I think the paper would need to be re-reviewed to ensure the findings still hold.

---

### Official Review · Reviewer_ZCbc · 2023-01-20
**n/a**

**Rating:** 4
**Confidence:** 4

**Review:**

Summary: This paper shows the impact of In Situ visual feedback on learning gesture commands in VR. The authors provided 4 different types of visual feedback: color, arrow, puppet, and ghost. They conducted two studies: Study 1 focuses on the design of In Situ feedback and Study 2 focuses on the precision of gestures and the autocompletion feature. The work finds that the novel feedback results in the most precise production of gestures.

Pros: The work explores the design of in-situ feedback for hand gestures in XR, following the findings of how to improve keyboard gesture learning

Cons: The work seems disorganized in the order of its sections and lacks clarity in the presentation of its motivation, methodology, and results. Impact of the work is unclear in part due to the structural issues in the paper. The text requires major revisions. The discussion and conclusion are minimalistic and do not sell the novelty of the paper.

Quality: Claims in Section 4.1 regarding how vital in situ feedback reads like an overstatement. This argument should make it clearer that the spatiality of the feedback is important, not specifically the types of In Situ feedback analyzed in this paper which do not synergize with the keyboard metaphor. Several aspects of the methodology are not well justified, e.g., why 99 gestures were included in Study 1 and why random selection was employed. It seems more useful to have a smaller set of realistic gestures and use counterbalancing to explore the space more thoroughly. Metrics such as “Enabled Task” are not defined

Clarity: The text is somewhat ambiguous regarding whether the gesture feedback is intended to occur all the time or just during a learning phase Section 4.1 reads like part of the motivation, so it feels late in the paper. Similarly, Section 4.3 seems like it should be part of Section 2. Section 3 should also be a subsection of Section 2 Paper is a bit repetitive, defining crib sheet again in Section 6.1 The numbers in Figure 5 are hard to read Was the pre-survey different between Study 1 and 2? The examples for manual dexterity are different

Originality: Originality of the work is limited. The main novelty is the design of the puppet in-situ feedback.

Significance: In situ feedback has the potential to improve the learnability of hand gestures in XR, but this impact is not well explored. The results provide data to inform the design feedback systems for hand gesture systems, but these findings are not focused on in the discussion or conclusion.

---

### Meta-Review · Area_Chair_HKR3 · 2023-01-20

**Recommendation:** 4
**Confidence:** 4

**Metareview:**

This paper presents the results from two studies that investigate how four types of visual feedback help people learn hand gestures in VR. The research question is interesting and relevant to the GI community. Even more so, the work intersects work in graphics and HCI, making it a very good fir for the conference. The structure of the paper makes sense and it is quite easy to follow. Reviwers do appreciate the amount of work that went into the design, implementation and analysis of two full studies.
That being said, there are quite a few issues with the submission, that in the opinion of all reviewers require a major revision of the work before it can be accepted at a conference like GI. My recommendation is that the authors address the weaknesses of the paper in a revised manuscript and resubmit the work, possible to the GI 2023 second deadline.

One key concern from reviewers is the lack of rationale for many design choices that have been made. While it does not invalidate the studies, providing these details would provide an additional layer of information that reviewers would need to review. This lack of rationale concerns several parts of the paper, specifically: the choice of the gesture set; the choice of these visual feedbacks; study choices (like the different hand poses that are studied, the metrics that are used for e.g., error, the thresholds used, etc..).

Another concern (that should be relatively easy to address) is that the study needs additional details. It relates a bit to the first point about rationale, in that several aspects need to be further elaborated upon. For example, the different statistics that were or were not run on the data, the choices for error rate ranges, the reason for using specific statistical tests, and better reporting of the results (it is actually not explained what the result figures with error basr represent - confidence intervals? at which level? standard deviation?).

Reviewers found that the paper lacks clarity and/or depth in several sections (motivation, methodology, results, impact, discussion/conclusion).

Finally, the writing needs to be carefully revised. The paper contains a good number of typos that should be easy to fix.

Reviewers provide thorough feedback with specific suggestions that I strongly encourage the authors to consider in reworking their manuscript for a future submission.